# Research on Fine-Tuning Optimization Strategies for Large Language Models in Tabular Data Processing

**DOI:** 10.3390/biomimetics9110708

**Published:** 2024-11-19

**Authors:** Xiaoyong Zhao, Xingxin Leng, Lei Wang, Ningning Wang

**Affiliations:** 1School of Management Science and Engineering, Beijing Information Science and Technology University, Beijing 100192, China; zhaoxiaoyong@bistu.edu.cn (X.Z.); wanglei575882@163.com (L.W.); wangningning@bistu.edu.cn (N.W.); 2Computer School, Beijing Information Science and Technology University, Beijing 100192, China

**Keywords:** data preprocessing, data noise, fine-tuning, generalization ability, large language models, model robustness, network security, tabular data

## Abstract

Recent advancements in natural language processing (NLP) have been significantly driven by the development of large language models (LLMs). Despite their impressive performance across various language tasks, these models still encounter challenges when processing tabular data. This study investigates the optimization of fine-tuning strategies for LLMs specifically in the context of tabular data processing. The focus is on the effects of decimal truncation, multi-dataset mixing, and the ordering of JSON key–value pairs on model performance. Experimental results indicate that decimal truncation reduces data noise, thereby enhancing the model’s learning efficiency. Additionally, multi-dataset mixing improves the model’s generalization and stability, while the random shuffling of key–value pair orders increases the model’s adaptability to changes in data structure. These findings underscore the significant impact of these strategies on model performance and robustness. The research provides novel insights into improving the practical effectiveness of LLMs and offers effective data processing methods for researchers in related fields. By thoroughly analyzing these strategies, this study aims to establish theoretical foundations and practical guidance for the future optimization of LLMs across a broader range of application scenarios.

## 1. Introduction

LLMs [1] have recently emerged as a key force in driving advancements in NLP [2] within artificial intelligence (AI) and machine learning (ML) research. These models excel in a variety of language tasks, including text generation, translation, and question-answering systems [3]. However, despite their strong performance across many applications, they encounter challenges when processing certain data types, particularly tabular data. This type of data is crucial across multiple domains, including cybersecurity, financial analysis, and medical diagnostics [4]. Therefore, the effective processing and optimization of tabular data has become an important research focus.

Tabular data [5], commonly stored in formats such as Comma-Separated Values (CSVs), Excel Spreadsheet (XLSX), Tab-Separated Values (TSVs), and OpenDocument Spreadsheet (ODS), are highly structured but still present challenges when processed by LLMs. The inherent format and characteristics of tabular data can make it difficult for these models to fully interpret and leverage their intrinsic structure, potentially hindering performance. Addressing these challenges through appropriate data preprocessing strategies, particularly in optimizing dataset construction and fine-tuning processes, is essential for enhancing model performance.

This study aims to optimize fine-tuning strategies for LLMs in the context of tabular data processing, focusing on three key aspects. First, we propose a data preprocessing method based on decimal truncation to reduce the impact of data noise [6] on model training. We validate the effectiveness of this method by comparing model performance before and after its application. Second, we explore multi-dataset mixing strategies to enhance model generalization, hypothesizing that combining data from diverse sources will improve consistency in model performance across various data distributions [7]. Finally, we investigate the impact of randomly shuffling key–value pair orders in JSON-formatted data [8] on model robustness. This aims to understand the model’s sensitivity to changes in data order and propose methods to improve robustness.

The significance of this research lies in its potential to substantially enhance the performance of LLMs in processing tabular data through optimized data preprocessing and fine-tuning strategies. This work not only provides new insights into improving the practical effectiveness of LLMs but also offers valuable data processing methods and strategies for researchers in related fields. Through a detailed analysis of these strategies, the study aims to provide both theoretical foundations and practical guidance for the future optimization of LLMs across a broader range of application scenarios.

The structure of this paper is as follows: Section 2 presents a comprehensive literature review, examining the existing research on noise handling, data integration, and sequence processing in AI and ML, and identifies the gaps that this study seeks to address. Section 3 outlines the methods employed, including the proposed data preprocessing strategies—decimal truncation, multi-dataset mixing, and randomization of key–value pair order—along with their corresponding implementation algorithms. Section 4 details the experimental setup, describing the datasets, data volume, evaluation metrics, and the design of Experiments A, B, and C. Section 5 reports the results of these experiments, highlighting the impact of the proposed strategies on model performance. Section 6 discusses the findings, their implications, and the study’s limitations. Finally, Section 7 concludes the paper by summarizing the key contributions and suggesting directions for future research.

## 2. Literature Review

Extensive research has advanced AI and ML models in areas such as noise handling, data integration, and sequence processing, which significantly impact model performance. However, certain gaps persist, particularly in applying these techniques to LLMs for tabular data processing.

Noisy data present a significant challenge in ML, often leading to overfitting and reduced accuracy. Nasution et al. [9] emphasize that feature reduction enhances decision tree classification accuracy, while Kavitha et al. [10] propose an ensemble framework that combines noise filtering with classification methods to effectively manage data irregularities. Xiao et al. [11] offer a comprehensive review of noise-handling methods in electronic health records, underscoring the importance of noise management in data-intensive fields. While noise reduction techniques are widely studied, the effects of decimal truncation on LLMs fine-tuning remain under-researched. This study addresses this gap by examining decimal truncation’s effectiveness in reducing noise and improving model accuracy.

The integration of multiple datasets to enhance model generalization has also garnered attention. Studies indicate that dataset mixing can improve robustness and generalization as shown by Liang et al. [12] in image classification. Zhu et al. [13] extend this approach to environmental data, demonstrating the benefits of adaptive fusion in heterogeneous contexts. However, research on dataset mixing for LLMs, especially in domains like cybersecurity where robust generalization is critical, remains limited. This study contributes to this area by evaluating various dataset mixing strategies to improve LLMs generalization on tabular data.

Data order has also been shown to influence model performance. The Transformer model introduced by Vaswani et al. [14] employs positional encoding, critical for handling sequence data. Chang et al. [15] further demonstrate that sample order impacts the training efficiency and output quality of neural data-to-text models. However, the effects of key–value pair order in JSON-format tabular data on model robustness are underexplored. This paper addresses this gap by systematically investigating the impact of shuffling key–value pairs in JSON data on LLMs robustness and proposing strategies to mitigate any negative effects.

Foundational models, such as BERT [16] and GPT-3 [17], have shown that pretrained LLMs are highly adaptable across various tasks. Emerging models like ELECTRA [18] have introduced efficiencies by employing discriminators instead of generators, which further informs our focus on optimizing LLMs for structured data processing through novel fine-tuning strategies.

In summary, while advancements have been made in noise handling, dataset mixing, and sequence order processing, specific applications in LLMs fine-tuning for tabular data remain underexplored. By addressing these gaps, this study aims to develop optimized strategies to enhance LLMs performance on tabular data, contributing novel insights to the broader field of AI and ML optimization.

## 3. Methods

This study introduces three data preprocessing strategies: decimal truncation, multi-dataset mixing, and randomizing key–value pair order. These strategies aim to enhance the performance of LLMs when processing complex data.

### 3.1. Decimal Truncation

The decimal truncation method standardizes and reduces numerical precision in JSON-formatted data by rounding decimal values to two decimal places. This preprocessing technique minimizes noise from excessive decimal precision, which is often irrelevant for downstream tasks. By doing so, it allows the model to focus on essential data features. Decimal truncation also reduces computational complexity and improves model stability, particularly when dealing with datasets that exhibit high precision variability across entries. The process involves the following steps:

Type Checking and Conversion: Each value in the JSON data is examined to determine if it is a decimal (i.e., a floating-point number). Non-decimal data types, such as integers, strings, and Booleans, are left unchanged, preserving their original values for categorical and non-numeric attributes.

Precision Reduction: If the value is identified as a decimal, it is truncated to two decimal places using standard rounding conventions. For example, a value of 169.3636364 would be rounded to 169.36. This truncation preserves the core information while removing less significant precision that could introduce noise during model training.

Conversion of Whole Decimal Values: After rounding, if the decimal value has no fractional part (e.g., 1.00), it is converted into an integer (e.g., 1). This step standardizes numerical representations across the dataset and optimizes storage, ensuring consistency, especially in datasets with diverse numerical fields.

Handling Special Cases: For fields where higher precision is critical (e.g., scientific data), additional decimal places may be retained. However, in this study, a two-decimal rounding policy is considered sufficient for the cybersecurity metrics dataset.

The implementation of the decimal truncation method is detailed in the following Algorithm 1, which outlines the steps for processing each value within the JSON structure:
**Algorithm 1** Rounding numerical values in data.**Require:** Input data structure *data* (a float)**Ensure:** Data with rounded numerical values  1: Define a function round_numbers(data)  2: **if** *data* is a float **then**  3:    Round the float value to two decimal places  4:    **if** the rounded value has no decimal portion **then**  5:      Convert the result to an integer type  6:    **end if**  7: **end if**  8: **return** Processed *data*

This algorithm illustrates the process of evaluating each data entry, applying rounding as needed, and converting values to integers when applicable. By implementing decimal truncation, we reduce unnecessary variability in the numerical data, allowing the model to focus on meaningful patterns without the distraction of irrelevant precision noise.

### 3.2. Multi-Dataset Mixing Strategy

The multi-dataset mixing strategy involves constructing a new dataset by randomly sampling data from multiple distinct datasets. This approach enhances the generalization capability of the model by integrating diverse data sources. To mitigate the risk of overfitting to any specific dataset, the newly formed dataset is shuffled to randomize the order of its entries. By incorporating data variability from various sources, this strategy helps prevent the model from learning dataset-specific patterns that may not generalize well to unseen data. The following Algorithm 2 outlines the implementation of this strategy:
**Algorithm 2** Multi-dataset mixing strategy.**Require:** List of file paths *sample_file_paths* containing data in JSON format; *sample_size* indicating the number of samples to select from each file**Ensure:** Mixed dataset with samples from all files in random order  1: Initialize an empty list *sampled_data*  2: **for** each *file_path* in *sample_file_paths* **do**  3:    Load data from *file_path*  4:    Randomly select *sample_size* samples from the loaded data  5:    Append the selected samples to *sampled_data*  6: **end for**  7: Randomly shuffle the entries in *sampled_data*  8: **return** Mixed dataset *sampled_data*

This method enhances the model’s robustness by combining data from various sources, thus increasing the diversity of the training set. By randomizing both the data selection and entry order, the model is less likely to memorize idiosyncratic features of individual datasets, improving its generalization ability. Furthermore, the strategy can be adapted to dynamically adjust the sample size based on the characteristics of each dataset or to weight datasets according to their relevance to the task, further optimizing the model’s learning process.

### 3.3. Randomizing Key–Value Pair Order

The randomizing key–value pair order method aims to shuffle the order of key–value pairs within each JSON data entry while preserving their inherent relationships. Initially, the key–value pair information is maintained, and the order of the keys is randomized. The corresponding values for each key are then identified, ensuring that the relationship between keys and their values remains intact. This process eliminates any potential biases introduced by key order, while retaining the structure of the original data. For example, the key–value pairs {“duration”: 0, “protocol_type”: “tcp”, “service”: “http”, “flag”: “SF”} might be shuffled to {“protocol_type”: “tcp”, “flag”: “SF”, “service”: “http”, “duration”: 0}. This randomization reduces the model’s dependency on the order of features, encouraging it to learn more generalized patterns.

The following Algorithm 3 outlines the implementation of this randomization strategy.
**Algorithm 3** Randomizing key–value pairs in data structure.**Require:** List of items *data_list* containing dictionaries with a sub-dictionary *data***Ensure:** Each *data* section within items is reordered with shuffled keys  1: **for** each *item* in *data_list* **do**  2:    Extract *data* from *item*  3:    Retrieve the list of keys from *data* and shuffle them randomly  4:    Reconstruct *data* using the shuffled keys and original values  5:    Replace the original *data* in *item* with *shuffled_data*  6: **end for**  7: **return** Processed *data_list*

This approach strengthens the model’s robustness by reducing overfitting to specific key orders, which can introduce learning biases. Randomizing the key order encourages the model to focus on the relationships between keys and values rather than memorizing their sequence. Furthermore, this method can be adapted to handle additional complexities, such as prioritizing specific key–value pairs based on their relevance, or implementing different randomization techniques depending on the characteristics of the data. Ultimately, the goal is to enhance the model’s ability to generalize across varying data orders, leading to more stable and consistent performance during testing.

## 4. Experiments

### 4.1. Datasets

This experiment utilized three widely used cybersecurity datasets: KDDCup’99 (a classic dataset for detecting network intrusions), UNSW-NB15 (a dataset incorporating recent network attacks) [19,20,21,22,23], and CICIDS2017 (a high-quality dataset simulating modern network traffic and attacks) [24]. To ensure consistency and data quality, the following preprocessing steps were applied:

Handling Missing Values: Records containing null values were removed to maintain data integrity.

Duplicate Handling: Duplicate records were deleted to prevent redundancy from affecting model training.

Balanced Sampling: Random sampling was performed on both normal and abnormal data to ensure equal proportions, thereby avoiding the impact of class imbalance on model performance.

Format Conversion: The original CSV-format datasets were converted to JSON format to meet the experimental requirements.

All subsequent experiments were conducted using these preprocessed datasets.

### 4.2. Data Volume

Relevant studies [25] indicate that performance improvement in fine-tuning LLMs gradually plateaus as the data volume reaches 1000 samples. This suggests that using 1000 samples for fine-tuning can yield significant performance gains, with diminishing returns from further data increases. Given the limited experimental resources, this experiment used 1000 fine-tuning samples and 300 test samples per dataset to ensure the model’s generalization ability. For mixed datasets, 3000 fine-tuning samples and 900 test samples were used to comprehensively evaluate the model’s performance across different data distributions.

### 4.3. Evaluation Metrics

To compare the effects of different strategies on the model, five evaluation metrics were employed: accuracy, precision, recall, F1 score [26], and a newly introduced metric, “1 − Range”, for assessing model stability.

Accuracy: Measures the overall classification performance by calculating the proportion of correctly predicted samples out of the total, reflecting the model’s general effectiveness.

Precision: Assesses the model’s ability to correctly identify positive samples, calculated as the proportion of true positives among all samples predicted as positive, indicating the model’s precision.

Recall: Evaluates the model’s ability to detect positive samples, measured as the proportion of actual positives correctly identified by the model, reflecting its detection capability.

F1 Score: Combines precision and recall into their harmonic mean, providing a balance between the model’s precision and detection ability.

1 − Range: This new metric, introduced in this experiment, measures model stability. While traditional metrics like accuracy, precision, recall, and F1 score evaluate overall performance, 1 − Range assesses the consistency of the model’s performance across different metrics. It is defined as the inverse of the difference between the maximum and minimum values across multiple evaluation metrics plus 1, with the formula 1 − Range = 1 − (Max − Min). A higher 1 − Range value indicates greater model stability.

The advantages of the 1 − Range metric include the following.

Consistency: 1 − Range aligns with the meaning and value range of other performance metrics, with higher values indicating better performance. Ease of Interpretation: The value of 1 − Range ranges from zero to one, making it easy to understand and interpret, with higher values denoting better stability.

High Adaptability: 1 − Range can assess model stability under different datasets or experimental conditions, offering researchers a simple and effective tool.

Theoretical Support: In statistics, range is a basic method for measuring data fluctuation [27]. Although less comprehensive than standard deviation or variance, its simplicity and intuitiveness make it useful for assessing model stability. 1 − Range retains the benefits of range while ensuring compatibility with traditional performance metrics.

### 4.4. Experimental Design

This experiment utilized the base model ‘google/gemma-2-2b-it’ [28] alongside LoRA [29] technology for fine-tuning. The experimental design followed the principle of controlled variables, ensuring that all parameters and settings remained constant when evaluating the impact of specific strategies. Detailed training parameters, such as learning rate and batch size, as well as additional experimental details, are provided in the Data Availability Statement section, along with relevant source codes and dataset links.

#### 4.4.1. Experiment A: Impact of Decimal Truncation on Fine-Tuning LLMs

Objective: To investigate whether decimal truncation improves the fine-tuning performance of LLMs.

Procedure:Data Preprocessing: Randomly sample data from the KDDCup’99, UNSW-NB15, and CICIDS2017 datasets, truncating the decimal places of numerical data to two digits (e.g., truncating 1.23456789 to 1.23).Model Training: Fine-tune the base model ‘google/gemma-2-2b-it’ using the truncated dataset.Control Experiment: Fine-tune the model using the original, untruncated dataset under the same conditions as the control group.Model Evaluation: Evaluate the performance of the models trained with both truncated and untruncated data using the same evaluation dataset.

Expected Outcome: Decimal truncation is anticipated to reduce irrelevant variations in the data, thereby lowering noise and enhancing the model’s learning efficiency. The model trained on truncated data is expected to demonstrate superior accuracy and robustness compared to the model trained on untruncated data.

#### 4.4.2. Experiment B: Impact of Multi-Dataset Mixing on the Generalization Ability of LLMs

Objective: To explore the effect of mixing datasets from different sources on the generalization ability of LLMs, particularly in fine-tuning performance on specific datasets.

Procedure:Data Preprocessing: Extract equal samples from the KDDCup’99, UNSW-NB15, and CICIDS2017 datasets to create four training sets. Training Sets A, B, and C consist of data from KDDCup’99, UNSW-NB15, and CICIDS2017, respectively, while Training Set D is a mixture of A, B, and C.Model Training: Fine-tune the base model ‘google/gemma-2-2b-it’ using Training Sets A, B, C, and D to obtain Models A, B, C, and D.Model Evaluation: Evaluate Models A and D using unseen KDDCup’99 data, Models B and D using unseen UNSW-NB15 data, and Models C and D using unseen CICIDS2017 data.

Expected Outcome: The mixed dataset, by incorporating data from different sources, is expected to cover a broader range of data distributions and feature patterns, providing richer information that enhances the model’s generalization ability. Consequently, Model D is expected to outperform Models A, B, and C across all three datasets.

#### 4.4.3. Experiment C: Impact of Randomizing Key–Value Pair Order on the Robustness of LLMs

Objective: To investigate whether randomizing the order of key–value pairs in JSON data improves the robustness of LLMs.

Procedure:Data Preprocessing: Extract samples from the KDDCup’99, UNSW-NB15, and CICIDS2017 datasets, mixing them to construct a standard dataset E. Randomly shuffle the order of key–value pairs in each JSON file to generate the experimental dataset F.Model Training: Fine-tune the base model ‘google/gemma-2-2b-it’ using datasets E and F.Model Evaluation: Test the fine-tuned models E and F using datasets with both ordered and shuffled key–value pairs. Compare the models’ robustness by evaluating their performance on ordered and randomized data using metrics such as 1 − Range.

Expected Outcome: Randomizing the order of key–value pairs in JSON data is expected to increase the dataset diversity and enhance the model’s robustness. Therefore, Model F should demonstrate greater stability and robustness when handling both ordered and randomized data.

## 5. Results

### 5.1. Experiment A: Impact of Decimal Truncation on Fine-Tuning LLMs

As illustrated in Figure 1 and Table 1, Experiment A evaluated the effects of decimal truncation on model fine-tuning. The findings revealed significant performance enhancements across multiple evaluation metrics for the model trained with truncated data. Notably, the truncated model demonstrated substantial improvements on the UNSW-NB15 dataset, increasing accuracy by 16.3%, recall by 42.4%, and F1 score by 26.7%. While the improvements on the CICIDS2017 dataset were less pronounced, recall and F1 score still rose by 9.1% and 1.4%, respectively. The KDDCup’99 dataset exhibited minimal changes, though there were slight gains in precision and the 1 − Range metric.

Overall, these results confirmed that decimal truncation effectively reduces noise in data, thereby enhancing the model’s learning efficiency. Moreover, the 1 − Range metric analysis showed that the truncated model exhibited superior stability, with a 13.8% increase in the 1 − Range value on the UNSW-NB15 dataset, indicating more consistent performance across different metrics.

### 5.2. Experiment B: Impact of Multi-Dataset Mixing on the Generalization Ability of LLMs

As shown in Figure 2 and Table 2, Experiment B assessed the impact of multi-dataset mixing on model generalization. The results indicate that models fine-tuned with mixed datasets outperformed those trained with single datasets across all metrics. Specifically, the mixed dataset model achieved a 5.0% improvement in accuracy, precision, and a 4.2% improvement in F1 score on the UNSW-NB15 dataset. Similar gains were observed on the CICIDS2017 dataset, with improvements of 4.3%, 7.4%, and 4.1%, respectively. On the KDDCup’99 dataset, accuracy, recall, and F1 score increased by 1.7%, 1.8%, and 1.6%, respectively.

Furthermore, the mixed dataset model outperformed single dataset models on the 1 − Range metric, particularly on the CICIDS2017 dataset, where the 1 − Range value increased by 5.4%. These findings suggest that mixed datasets offer richer and more diverse feature information, significantly enhancing the model’s generalization capability and stability.

### 5.3. Experiment C: Impact of Randomizing Key–Value Pair Order on the Robustness of LLMs

Figure 3 and Table 3 summarize the findings from Experiment C, which investigated the effect of randomizing JSON key–value pair orders on model robustness. The results revealed significant performance fluctuations for the model fine-tuned on ordered data when tested with shuffled key–value pairs. Specifically, when the model trained on ordered data was tested with shuffled data, the accuracy, recall, and F1 scores dropped dramatically, with decreases of 44.0%, 87.3%, and 79.4% on the CICIDS2017 dataset, and 27.7%, 62.7%, and 43.5% on the UNSW-NB15 dataset. This suggests that the model struggles to maintain stable performance when confronted with data sequences differing from its training order.

Conversely, the model fine-tuned with shuffled key–value pairs displayed better robustness when handling both ordered and shuffled data. For example, on the KDDCup’99 dataset, the model fine-tuned with shuffled data achieved F1 scores of 97.1% and 95.3% on ordered and shuffled data, respectively, indicating stronger adaptability. Additionally, the higher 1 − Range value for the shuffled model suggests more consistent performance across different metrics.

In conclusion, while randomizing key–value pair orders may lead to performance degradation in certain cases, it generally enhances the model’s adaptability to changes in key–value pair order, thereby improving robustness and stability.

### 5.4. Additional Experiment: Phi-3 Model Results

Figure 4 and Table 4 present results from an additional experiment conducted with the ‘microsoft/Phi-3-mini-4k-instruct’ [30] model to verify the generalizability of the previously discussed methods. These results further support the conclusions, demonstrating that techniques such as decimal truncation, multi-dataset mixing, and randomizing key–value pair order consistently yield effective results across different models.

## 6. Discussion

This study systematically evaluated the effects of data preprocessing techniques, dataset mixing strategies, and key–value pair ordering on the performance of LLMs through Experiments A, B, and C. The findings not only confirmed our hypotheses regarding data processing but also provided new insights into enhancing model generalization and robustness.

### 6.1. Impact of Decimal Truncation on Model Fine-Tuning (Experiment A)

Experiment A demonstrated that decimal truncation significantly enhances model performance across various evaluation metrics, validating its effectiveness in reducing data noise. By minimizing extraneous details in the input data, decimal truncation reduces computational complexity, allowing the model to focus on essential features while ignoring noise-inducing elements, thereby improving overall accuracy. This approach parallels dimensionality reduction techniques [31], which enhance model learning by eliminating unnecessary information. However, decimal truncation may lead to information loss, particularly in tasks requiring high-precision numerical data, potentially diminishing the model’s fine-grained predictive capabilities. Future research should explore optimized truncation strategies to balance information preservation and noise reduction effectively.

### 6.2. Impact of Multi-Dataset Mixing on Model Generalization (Experiment B)

Experiment B revealed that fine-tuning with mixed datasets significantly improves the model’s generalization ability. Models trained on mixed datasets outperformed those trained on single datasets across all metrics, consistent with diversity learning theory. This theory suggests that integrating data from diverse sources enables models to learn a broader range of features, thereby enhancing generalization [32]. Theoretically, mixing multiple datasets increases sample and feature diversity during training, helping the model learn from a wider array of patterns [33] and reducing the risk of overfitting. Additionally, differences between datasets encourage the model to develop more robust feature representations, leading to stable performance across varying data distributions [34]. Future research should investigate the impact of dataset similarity on model performance to optimize dataset mixing strategies further.

### 6.3. Impact of Randomizing Key–Value Pair Order on Model Robustness (Experiment C)

Experiment C explored the impact of key–value pair order on model robustness, revealing that models are highly sensitive to changes in this order. When trained on ordered data and tested on randomly ordered data, model performance declined significantly. This suggests that LLMs have poor robustness when the data order is critical, particularly when there is a mismatch between the training and testing data order. These models rely on the sequence information of input data to learn patterns and relationships; if this order is disrupted during testing, the model may fail to interpret the data structure, leading to performance degradation [35]. Conversely, models fine-tuned with randomly ordered data demonstrated more stable performance across different data orders, suggesting that this approach helps the model learn more generalized feature representations, reducing dependency on specific data order. Future research could introduce sequence randomization strategies or develop enhanced model architectures to improve robustness against order-dependent data.

### 6.4. Limitations and Future Directions

Despite significant progress in improving LLMs performance, this study has limitations, and several areas require further exploration.

First, the experiments were conducted on only three datasets and two models. While these datasets are representative within the cybersecurity domain, the generalizability of the results to other fields, tasks, and models remains unvalidated. Future research should expand to include a wider variety of datasets, tasks, and models to ensure the broad applicability of these methods.

Second, the decimal truncation and key–value pair randomization strategies used may not be universally suitable for all data types. In the case of decimal truncation, there is a risk of information loss, particularly in tasks requiring high-precision numerical information. Future research should explore more intelligent truncation strategies that aim to reduce noise while preserving critical information.

Enhancing model robustness to changes in input order is another critical research direction. Future studies should investigate order-independent preprocessing methods or design enhanced model architectures to improve adaptability to order-independent data.

Finally, there is substantial potential for optimizing multi-dataset mixing strategies. For instance, dynamically adjusting the mixing ratio of datasets or adaptively selecting datasets based on task characteristics could further enhance model performance in specific domains. In-depth research into these strategies could provide more effective improvements for models across various applications.

## 7. Conclusions

This study systematically investigated the effects of data preprocessing strategies, dataset mixing methods, and key–value pair order on the performance of LLMs, yielding several key findings and contributions:

Experiment A confirmed that decimal truncation as a data preprocessing technique significantly enhances model performance across various evaluation metrics. This improvement is particularly pronounced in datasets with high noise levels, suggesting that reducing irrelevant details in the data optimizes the model’s learning efficacy. Although decimal truncation may lead to some information loss, it effectively improves model stability and accuracy in practical applications, especially when handling noisy data. This method presents a valuable approach for large-scale data processing in real-world scenarios.

Experiment B demonstrated that multi-dataset mixing strategies significantly enhance the model’s generalization ability. By integrating data from diverse sources, the model exhibited stronger robustness and consistency across various testing environments. This strategy has broad applicability beyond cybersecurity, extending to fields such as financial risk control and medical diagnosis, where data diversity and generalization are crucial. As more diverse datasets are introduced and model training techniques evolve, this approach is expected to further advance the application of LLMs in complex scenarios.

Experiment C revealed that LLMs are highly sensitive to input order, highlighting their limitations in handling order-independent data. This finding emphasizes the need to improve the robustness of current model architectures when dealing with inconsistent input sequences. This has important implications for practical applications, particularly in scenarios where data order is irrelevant or inconsistent, such as natural language understanding, question-answering systems, and tabular data analysis. Future research should explore more advanced model architectures or introduce order-independent preprocessing methods to enhance adaptability to these data types.

The results of this study hold significant theoretical importance and offer broad potential for practical applications. As LLMs become increasingly integrated across various fields, optimizing data processing strategies will directly impact their effectiveness and user experience. Future research should continue to refine data processing strategies in different application scenarios, thereby promoting the widespread use of LLMs in more complex and dynamic environments.

In conclusion, this study provides a set of effective strategies for optimizing the performance of LLMs, particularly in addressing noisy data, enhancing generalization ability, and improving robustness. We anticipate further application and validation of these methods across various fields, offering more intelligent and precise technical support to all levels of society.

## Figures and Tables

**Figure 1 biomimetics-09-00708-f001:**
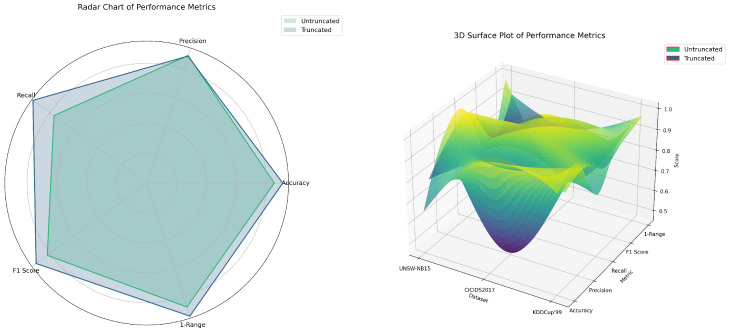
Performance comparison of models with and without decimal truncation across different datasets.

**Figure 2 biomimetics-09-00708-f002:**
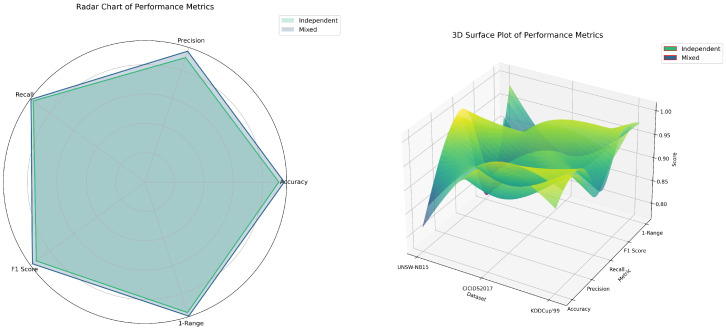
Impact of multi-dataset mixing on model generalization across various datasets.

**Figure 3 biomimetics-09-00708-f003:**
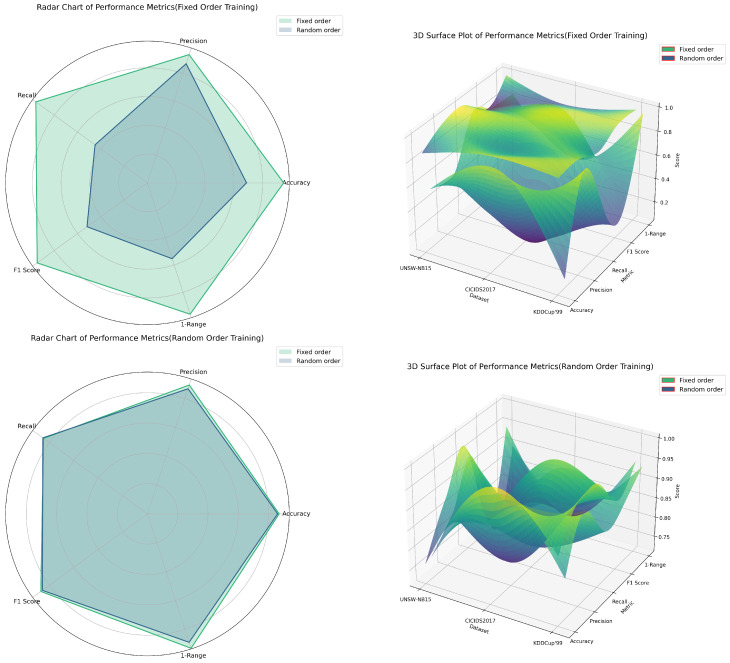
Comparative analysis of fixed vs. random key–value pair orders on model performance.

**Figure 4 biomimetics-09-00708-f004:**
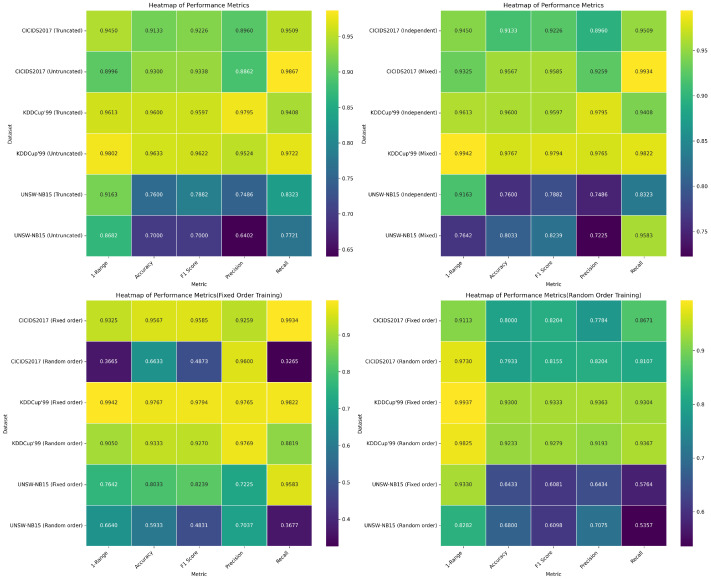
Evaluation of Phi-3 model performance with various data preprocessing techniques.

**Table 1 biomimetics-09-00708-t001:** Quantitative impact of decimal truncation on model performance.

Dataset	Accuracy	Precision	Recall	F1 Score	1 − Range
UNSW-NB15 (Untruncated)	0.6533	0.7097	0.4615	0.5593	0.7519
CICIDS2017 (Untruncated)	0.9300	0.9783	0.8824	0.9278	0.9041
KDDCup’99 (Untruncated)	**0.9767**	**1.0000**	**0.9551**	**0.9770**	**0.9551**
UNSW-NB15 (Truncated)	0.8167	0.7751	0.8851	0.8265	0.8900
CICIDS2017 (Truncated)	0.9400	0.9125	**0.9733**	0.9419	0.9392
KDDCup’99 (Truncated)	**0.9733**	**0.9866**	0.9608	**0.9735**	**0.9742**

*Note:* Bold values indicate the best performance for each metric.

**Table 2 biomimetics-09-00708-t002:** Improvement in model generalization through multi-dataset mixing.

Training Dataset	Testing Dataset	Accuracy	Precision	Recall	F1 Score	1 − Range
UNSW-NB15	UNSW-NB15	0.8167	0.7751	0.8851	0.8265	0.8900
CICIDS2017	CICIDS2017	0.9400	0.9125	**0.9733**	0.9419	0.9392
KDDCup’99	KDDCup’99	**0.9733**	**0.9866**	0.9608	**0.9735**	**0.9742**
Mixed	UNSW-NB15	0.8667	0.8250	0.9167	0.8684	0.9083
Mixed	CICIDS2017	0.9833	0.9864	**0.9797**	0.9831	**0.9933**
Mixed	KDDCup’99	**0.9900**	**1.0000**	0.9786	**0.9892**	0.9786

*Note:* Bold values indicate the best performance for each metric.

**Table 3 biomimetics-09-00708-t003:** Effect of key–value pair order variation on model robustness.

Training Dataset	Testing Dataset	Accuracy	Precision	Recall	F1 Score	1 − Range
Fixed order	UNSW-NB15 (Fixed order)	0.8667	0.8250	0.9167	0.8684	0.9083
Fixed order	CICIDS2017 (Fixed order)	0.9833	0.9864	**0.9797**	0.9831	**0.9933**
Fixed order	KDDCup’99 (Fixed order)	**0.9900**	**1.0000**	0.9786	**0.9892**	0.9786
Fixed order	UNSW-NB15 (Random order)	0.5900	0.8545	0.2901	0.4332	0.4356
Fixed order	CICIDS2017 (Random order)	0.5433	0.8421	0.1067	0.1893	0.2646
Fixed order	KDDCup’99 (Random order)	**0.9333**	**0.9156**	**0.9527**	**0.9338**	**0.9629**
Random order	UNSW-NB15 (Fixed order)	0.7600	0.7400	0.7708	0.7551	**0.9692**
Random order	CICIDS2017 (Fixed order)	0.8800	0.9444	0.8293	0.8831	0.8848
Random order	KDDCup’99 (Fixed order)	**0.9700**	**1.0000**	**0.9430**	**0.9707**	0.9430
Random order	UNSW-NB15 (Random order)	0.7833	0.7195	0.8613	0.7841	0.8582
Random order	CICIDS2017 (Random order)	0.8467	0.8992	0.7785	0.8345	0.8793
Random order	KDDCup’99 (Random order)	**0.9500**	**0.9870**	**0.9212**	**0.9530**	**0.9342**

*Note:* Bold values indicate the best performance for each metric.

**Table 4 biomimetics-09-00708-t004:** Performance evaluation of Phi-3 model with different data preprocessing techniques.

Experiment	Training Dataset	Testing Dataset	Accuracy	Precision	Recall	F1 Score	1 − Range
A	UNSW-NB15 (Untruncated)	UNSW-NB15 (Untruncated)	0.7000	0.6402	0.7721	0.7000	0.8682
A	CICIDS2017 (Untruncated)	CICIDS2017 (Untruncated)	0.9300	0.8862	**0.9867**	0.9338	0.8996
A	KDDCup’99 (Untruncated)	KDDCup’99 (Untruncated)	**0.9633**	**0.9524**	0.9722	**0.9622**	**0.9802**
A, B	UNSW-NB15 (Truncated)	UNSW-NB15 (Truncated)	0.7600	0.7486	0.8323	0.7882	0.9163
A, B	CICIDS2017 (Truncated)	CICIDS2017 (Truncated)	0.9133	0.8960	**0.9509**	0.9226	0.9450
A, B	KDDCup’99 (Truncated)	KDDCup’99 (Truncated)	**0.9600**	**0.9795**	0.9408	**0.9597**	**0.9613**
B, C	Mixed (Fixed order)	UNSW-NB15 (Fixed order)	0.8033	0.7225	0.9583	0.8239	0.7642
B, C	Mixed (Fixed order)	CICIDS2017 (Fixed order)	0.9567	0.9259	**0.9934**	0.9585	0.9325
B, C	Mixed (Fixed order)	KDDCup’99 (Fixed order)	**0.9767**	**0.9765**	0.9822	**0.9794**	**0.9942**
C	Mixed (Fixed order)	UNSW-NB15 (Random order)	0.5933	0.7037	0.3677	0.4831	0.6640
C	Mixed (Fixed order)	CICIDS2017 (Random order)	0.6633	0.9600	0.3265	0.4873	0.3665
C	Mixed (Fixed order)	KDDCup’99 (Random order)	**0.9333**	**0.9769**	**0.8819**	**0.9270**	**0.9050**
C	Mixed (Random order)	UNSW-NB15 (Fixed order)	0.6433	0.6434	0.5764	0.6081	0.9330
C	Mixed (Random order)	CICIDS2017 (Fixed order)	0.8000	0.7784	0.8671	0.8204	0.9113
C	Mixed (Random order)	KDDCup’99 (Fixed order)	**0.9300**	**0.9363**	**0.9304**	**0.9333**	**0.9937**
C	Mixed (Random order)	UNSW-NB15 (Random order)	0.6800	0.7075	0.5357	0.6098	0.8282
C	Mixed (Random order)	CICIDS2017 (Random order)	0.7933	0.8204	0.8107	0.8155	0.9730
C	Mixed (Random order)	KDDCup’99 (Random order)	**0.9233**	**0.9193**	**0.9367**	**0.9279**	**0.9825**

*Note:* Bold values indicate the best performance for each metric.

## Data Availability

Code for fine-tuning the model: https://www.kaggle.com/code/lengxingxin/code-for-fine-tuning (accessed on 5 November 2024); Code for testing the model: https://www.kaggle.com/code/lengxingxin/code-for-testing (accessed on 5 November 2024); Preprocessed dataset UNSW-NB15: https://www.kaggle.com/datasets/lengxingxin/unsw-nb15-20000-json (accessed on 5 November 2024); Preprocessed dataset CICIDS2017: https://www.kaggle.com/datasets/lengxingxin/cicids2017-20000-json (accessed on 5 November 2024); Preprocessed dataset KDDCup’99: https://www.kaggle.com/datasets/lengxingxin/kdd99-20000-json (accessed on 5 November 2024).

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
