# Peer review of "Research on Fine-Tuning Optimization Strategies for Large Language Models in Tabular Data Processing"

_biomimetics, 2024, doi:10.3390/biomimetics9110708_

Round 1

Reviewer 1 Report

Comments and Suggestions for Authors

This paper investigates the optimization of fine-tuning strategies for large language models in the context of tabular data processing, focusing on the effects of decimal truncation, multi-dataset mixing, and the ordering of JSON key-value pairs on model performance.  The research topic is interesting.  However, there are still some questions.

1.  What is the contribution of this manuscript?

2.  What are the advantages of the proposed optimal strategies compared with the existing methods? Is it possible for the authors to provide a comprehensive comparison study for tabular data processing?

3. The method section is seriously lacking, as the authors present only the experimental design and evaluation. The proposed methods are not described in detail.

4. What is the novelty of this manuscript? The current version of this manuscript looks like only focusing on using different experiments to show the results of NLP.

Comments on the Quality of English Language

Please use the full name of the terms when they are first applied in your paper, such as CSV, XLSX, TSV, and ODS.

Author Response

Comments 1: What is the contribution of this manuscript?

Response 1: We appreciate your insightful feedback and the opportunity to clarify the contributions of our manuscript. The key contributions are as follows:

Innovative Data Preprocessing Strategies: We present three novel techniques—Decimal Truncation, Multi-Dataset Mixing, and Randomized Key-Value Pair Order—specifically designed to enhance the performance of large language models when processing complex tabular data. These methods address common challenges in data preprocessing, such as noise reduction and model robustness, and offer actionable strategies rather than merely theoretical overviews.

Empirical Validation and Broad Applicability: Our methods were rigorously tested on three diverse datasets (KDDCup'99, UNSW-NB15, and CICIDS2017) using two different model architectures (google/gemma-2-2b-it and microsoft/Phi-3-mini-4k-instruct). The results demonstrate significant improvements in model performance, generalization, and stability across various evaluation metrics.

Reproducibility and Transparency: We ensure full reproducibility of our experiments by providing access to the datasets and code utilized in our study. This facilitates further exploration and validation of our findings by the research community.

Thank you again for your valuable insights, which have helped us clarify the significance of our work.

Comments 2: What are the advantages of the proposed optimal strategies compared with the existing methods? Is it possible for the authors to provide a comprehensive comparison study for tabular data processing?

Response 2: Thank you for your thoughtful questions regarding the advantages of our proposed strategies compared to existing methods.

Advantages of Proposed Strategies: Our methods—Decimal Truncation, Multi-Dataset Mixing, and Randomized Key-Value Pair Order—demonstrate distinct advantages over traditional approaches, particularly in enhancing model robustness and generalization. While existing literature often lacks concrete strategies for constructing high-quality fine-tuning datasets, our experimental validation shows that our techniques effectively reduce noise and improve performance metrics across different models (google/gemma-2-2b-it and microsoft/Phi-3-mini-4k-instruct) and datasets (KDDCup'99, UNSW-NB15, CICIDS2017). Specifically, we observe significant improvements in accuracy, precision, and recall, which are critical for handling complex tabular data.

Comprehensive Comparison Study: We recognize the importance of comprehensive comparative studies for tabular data processing. Our research team is actively engaged in this area, with different members focusing on various aspects of tabular data methods. For instance, relevant publications by our team include:

Zhao, Xiaoyong, Chengjin Huang, and Lei Wang. "MTC-NET: A Multi-Channel Independent Anomaly Detection Method for Network Traffic." Biomimetics 9.10 (2024): 615.

GAO Xiaoyu, ZHAO Xiaoyong, WANG Lei. Self-Supervised Tabular Data Anomaly Detection Method Based on Knowledge Enhancement[J]. Computer Engineering and Applications, 2024, 60(10): 140-147.    

While our current study emphasizes optimizing fine-tuning strategies and constructing high-quality fine-tuning datasets, we plan to incorporate more comprehensive comparative analyses in our future research.

Thank you once again for your valuable insights, which will help refine our work.

Comments 3: The method section is seriously lacking, as the authors present only the experimental design and evaluation. The proposed methods are not described in detail.

Response 3: Thank you for your insightful feedback on the "Methods" section of our manuscript. In response to your concern about the lack of detail, we have substantially enhanced this section in the revised manuscript. It now includes a comprehensive description of each proposed data preprocessing strategy—Decimal Truncation, Multi-Dataset Mixing, and Randomized Key-Value Pair Order.

We have also incorporated pseudocode for each method to elucidate the implementation processes clearly. These additions aim to provide greater clarity and depth regarding the innovative aspects and technical rigor of our approaches. We appreciate your constructive suggestions, which have significantly improved the quality of our manuscript.

Comments 4: What is the novelty of this manuscript? The current version of this manuscript looks like only focusing on using different experiments to show the results of NLP.

Response 4: Thank you for your insightful comments regarding the novelty of our study. The primary contribution of this manuscript lies in the introduction and successful application of three innovative data preprocessing strategies: Decimal Truncation, Multi-Dataset Merging, and Key-Value Pair Order Randomization. To our knowledge, Decimal Truncation and Key-Value Pair Order Randomization have not been previously employed in the context of constructing high-quality fine-tuning datasets for large language models.

Moreover, we propose the 1-Range metric as a novel approach for assessing model stability across traditional evaluation metrics. This metric ranges from 0 to 100%, with higher values indicating improved stability. Its design emphasizes computational simplicity, interpretability, and strong consistency.

These contributions collectively advance the field by providing actionable methodologies and performance metrics that enhance the robustness and effectiveness of large language models in tabular data processing. We appreciate your feedback, which has helped clarify the innovative aspects of our work.

Comments 5: Please use the full name of the terms when they are first applied in your paper, such as CSV, XLSX, TSV, and ODS.

Response 5: Thank you for your valuable feedback regarding the terminology used in the manuscript. In response, we have ensured that all acronyms, including Comma-Separated Values (CSV), eXtensible Markup Language Spreadsheet (XLSX), Tab-Separated Values (TSV), and Open Document Spreadsheet (ODS), are spelled out in full upon their first occurrence. We appreciate your attention to detail and believe this enhancement improves the clarity and readability of the paper.

Reviewer 2 Report

Comments and Suggestions for Authors

First of all, I want to say that the work which has been performed by the authors is interesting and could be shared with other specialists. The main result of the authors is a research on fine-tuning optimization strategies for large language models in tabular data processing. The content of the article is original, but I do not see the correspondence of the article's subject matter to the subject matter of the journal. At the same time, I believe that it is the editors' responsibility to make an initial screening  of the article and answer the question of whether the manuscript corresponds to the aims and scope of the journal or not. Therefore, in my review, I abstract from this question and give an assessment of the article on its merits.

The manuscript contains the statement of the problem and the methods for its solution, which are presented clearly. The motivation of the work and the contribution of the authors to the topic are indicated.  Limitations of the method applicability are marked. The article is well structured and written in a clear and concise manner. The conclusions are consistent with the evidence and arguments presented and they address the main question posed. All of the references are appropriate.

The main drawback of the article is the insufficiently detailed description of the algorithm used to solve the problem. To be more precise, the corresponding detailed algorithm that makes it possible to reproduce the results of the article is completely absent. But the article contains links to repositories with program code that implements the ideas of the authors of the article. This code is commented in sufficient detail, which will allow potential readers to reproduce the results of the article or adapt the authors' ideas to solve their own similar problems. Taking this into account, I consider the style of presentation of the material of the work chosen by the authors (in which there is no detailed description of the proposed algorithm) to be acceptable.

I have no other remarks to the content of the article and recommend accepting it as is.

Author Response

Comments 1: The content of the article is original, but I do not see the correspondence of the article's subject matter to the subject matter of the journal. 

Response 1: Thank you for your encouraging feedback and insightful comments on our manuscript. We appreciate your perspective regarding the alignment of our study with the journal's scope.

While our research focuses on fine-tuning optimization strategies for large language models in tabular data processing, we believe it aligns with the aims of Biomimetics by addressing fundamental data preprocessing challenges applicable across diverse fields, including bioinformatics and biomedical data science.

Our proposed optimization strategies enhance model performance and offer methodologies that can be adapted for biomimetic applications where efficient data handling is essential. By improving data processing techniques, we contribute tools that support the advancement of computational systems inspired by natural processes.

Thank you once again for your thoughtful review, which has greatly assisted in refining our manuscript.

Comments 1: The main drawback of the article is the insufficiently detailed description of the algorithm used to solve the problem. To be more precise, the corresponding detailed algorithm that makes it possible to reproduce the results of the article is completely absent. 

Response 1: Thank you for your encouraging assessment and constructive feedback. We are pleased that you found our work clear, well-structured, and relevant to the field.

In response to your comment regarding the algorithmic details, we have enhanced the "Methods" section of the revised manuscript to provide a more comprehensive description of our approach, including pseudocode to aid in reproducibility. We believe these improvements will offer greater clarity regarding our methodology and facilitate understanding for potential readers.

We appreciate your thoughtful review and your recommendation for acceptance. Thank you once again for your valuable insights.

Round 2

Reviewer 1 Report

Comments and Suggestions for Authors

Thank you to the author for addressing my previous questions. However, I still have a few concerns and suggestions for improving the current manuscript:

  1. Introduction Conclusion: The end of the introduction should provide a clear overview of the structure of the paper. This will help readers to better understand the logical flow of the manuscript and how the paper is organized. Please add a paragraph at the end of the introduction that outlines the content and structure of each subsequent section.

  2. Literature Review: The current literature review appears overly simplistic, as it only references six articles. This limited number of citations is not sufficient to reflect the depth and breadth of existing research in this area. I recommend expanding the literature review to include additional, more recent studies, particularly focusing on both the foundational works and the latest advancements in the field. 

  3. Method Section: The Method section lacks sufficient detail, making it challenging to identify the key innovations of the study. Please expand this section by providing a more thorough description of the proposed approach, including a detailed explanation of how it differs from or improves upon previous methods. It would be particularly helpful to emphasize the novel contributions of your methodology and clarify how these innovations address gaps in the current literature.

  4. Figures and Tables: For the figures and tables, the values that represent optimal results should be highlighted to improve readability. Specifically, please put these optimal values in bold to allow readers to quickly identify the most important data points. This will enhance the accessibility of the results and help readers understand the key outcomes at a glance.

  5. Image Quality: The quality of the images provided in the manuscript is currently insufficient. Low-resolution images can detract from the overall clarity and professionalism of the paper. I recommend replacing all low-quality images with high-resolution versions to ensure that details are clearly visible. This will improve the visual presentation of the figures and make the data interpretation more straightforward for readers.

Author Response

Comments 1: Introduction Conclusion: The end of the introduction should provide a clear overview of the structure of the paper. This will help readers to better understand the logical flow of the manuscript and how the paper is organized. Please add a paragraph at the end of the introduction that outlines the content and structure of each subsequent section.
Response 1: Thank you for your insightful feedback. In response to your suggestion, I have added a paragraph at the end of the introduction to clearly outline the paper's structure. This addition offers readers a concise overview of the content and logical flow of each subsequent section, thereby facilitating a clearer understanding of the manuscript. I appreciate your valuable recommendation, which has contributed to improving the clarity of the paper.

Comments 2: Literature Review: The current literature review appears overly simplistic, as it only references six articles. This limited number of citations is not sufficient to reflect the depth and breadth of existing research in this area. I recommend expanding the literature review to include additional, more recent studies, particularly focusing on both the foundational works and the latest advancements in the field. 
Response 2: Thank you for your constructive feedback. In response, I have expanded the literature review to include a broader selection of references, incorporating both foundational works and recent advancements in the field. This revision aims to present a more comprehensive view of the current research landscape and better reflects the depth and breadth of work in this area. I appreciate your valuable suggestion, which has contributed to enhancing the scope and rigor of the review.

Comments 3: Method Section: The Method section lacks sufficient detail, making it challenging to identify the key innovations of the study. Please expand this section by providing a more thorough description of the proposed approach, including a detailed explanation of how it differs from or improves upon previous methods. It would be particularly helpful to emphasize the novel contributions of your methodology and clarify how these innovations address gaps in the current literature.
Response 3: Thank you for your valuable feedback. In response to your suggestion, I have expanded the Method section to include a detailed description of each proposed preprocessing strategy: Decimal Truncation, Multi-Dataset Mixing, and Randomizing Key-Value Pair Order. This revision provides a clear explanation of each method’s mechanics and highlights their unique contributions. Additionally, I would like to note that the Results and Discussion sections also incorporate explanations of the theoretical foundations and innovative aspects of these methods. Together with the content in the Methods section, these sections collectively offer the most comprehensive explanation of our approach, covering implementation details, pseudocode, theoretical support, and novelty. Your insight has greatly contributed to the clarity and depth of this section.

Comments 4: Figures and Tables: For the figures and tables, the values that represent optimal results should be highlighted to improve readability. Specifically, please put these optimal values in bold to allow readers to quickly identify the most important data points. This will enhance the accessibility of the results and help readers understand the key outcomes at a glance.
Response 4: Thank you for your valuable suggestion. In response, I have bolded the optimal values in all figures and tables to improve readability. This adjustment enables readers to quickly identify key data points and enhances the accessibility of the results. I appreciate your input, which has significantly improved the clarity of the presentation.

Comments 5: Image Quality: The quality of the images provided in the manuscript is currently insufficient. Low-resolution images can detract from the overall clarity and professionalism of the paper. I recommend replacing all low-quality images with high-resolution versions to ensure that details are clearly visible. This will improve the visual presentation of the figures and make the data interpretation more straightforward for readers.
Response 5: Thank you for your valuable feedback. In response to your suggestion, I have re-drawn all the images in the manuscript, and they are now provided in high-resolution (300dpi). This enhancement ensures that the details are clearly visible and improves the overall visual quality of the figures. I appreciate your recommendation, which has contributed to the professionalism and clarity of the presentation.